# Survival after surgery for spinal metastatic disease: a nationwide multiregistry cohort study

Christian Carrwik ![ORCID],[1] Claes Olerud,[1] Yohan Robinson ![ORCID] [1,2,3]

## ABSTRACT

**Objectives** To evaluate survival after surgery and indications for surgery due to spinal metastatic disease.

**Design** A retrospective longitudinal multiregistry nationwide cohort study.

**Setting** 19 public hospitals in Sweden with spine surgery service, where 6 university hospitals account for over 90% of the cases.

**Participants** 1820 patients 18 years or older undergoing surgery due to spinal metastatic disease 2006–2018 and registered in Swespine, the Swedish national spine surgery registry.

**Interventions** Decompressive and/or stabilising spine surgery due to spinal metastatic disease.

**Primary outcome** Survival (median and mean) after surgery.

**Secondary outcomes** Indications for surgery, types of surgery and causes of death.

**Results** The median estimated survival after surgery was 6.2 months (95% CI: 5.6 to 6.8) and the mean estimated survival time was 12.2 months (95% CI: 11.4 to 13.1). Neurologic deficit was the most common indication for surgery and posterior stabilisation was performed in 70.5% of the cases. A neoplasm was stated as the main cause of death for 97% of the patients.

**Conclusion** Both median and mean survival times were well above the generally accepted thresholds for surgical treatment for spinal metastases, suggesting that patient selection for surgical treatment on a national level is adequate. Further research on quality of life after surgery and prognostication is needed.

[1]Department of Surgical Sciences, Uppsala University, Uppsala, Sweden
[2]Department of Research and Development, Armed Forces Centre for Defence Medicine, Vastra Frolunda, Sweden
[3]Institute of Clinical Sciences, Sahlgrenska Academy, Gothenburg, Sweden

**Correspondence to**
Dr Christian Carrwik;
christian.carrwik@surgsci.uu.se

## Strengths and limitations of this study

► Large nationwide study with reliable data on demographic and survival.
► Includes data on indications, surgical methods and tumour types.
► Multiregistry study with linkage to cause of death register.
► High level of missing data on quality of life before and after surgery.
► No control group with patients treated only non-surgically.

justify the risks with surgery. On the other hand, no patient should be excluded from potentially beneficial surgery. This dilemma highlights the need for tools for decision-making in oncological spine surgery, and there is an abundance of scoring systems, developed to predict survival after surgery. Several commonly used scoring systems have problems estimating survival, which might withhold potentially beneficial surgery from patients.[3–5]

In recent years, mortality rates in several common types of cancer known to cause metastatic spine disease have decreased. This is especially evident in high-income countries, where a higher level of access to spine surgery could be expected.[6] A drawback of most scoring systems is that they are often based on retrospective studies of patients treated decades ago, not reflecting recent advancements in oncology.[7]

The main objective of this study was to study survival after surgery for metastatic spine disease stratified by tumour type in a nationwide cohort. Secondary objects of investigation were indications for surgery, types of surgery and causes of death.

## INTRODUCTION

While there is strong evidence that surgical treatment can improve quality of life for patients with metastatic spine disease, predicting outcome after surgery is still a matter of extensive research.[1] The patients with spinal metastases often present with acute symptoms such as progressive neurological impairment, giving limited time to evaluation and decision-making. While the surgery might have a positive effect on health-related quality of life (HRQoL), it is not intended to increase survival and is associated with complications.[2]

On the one hand, patients selected for surgery for spinal metastatic disease should have enough expected survival time to

## MATERIALS AND METHODS

### Study design

This is an observational multiregistry cohort study of adult patients undergoing surgery

BMJ

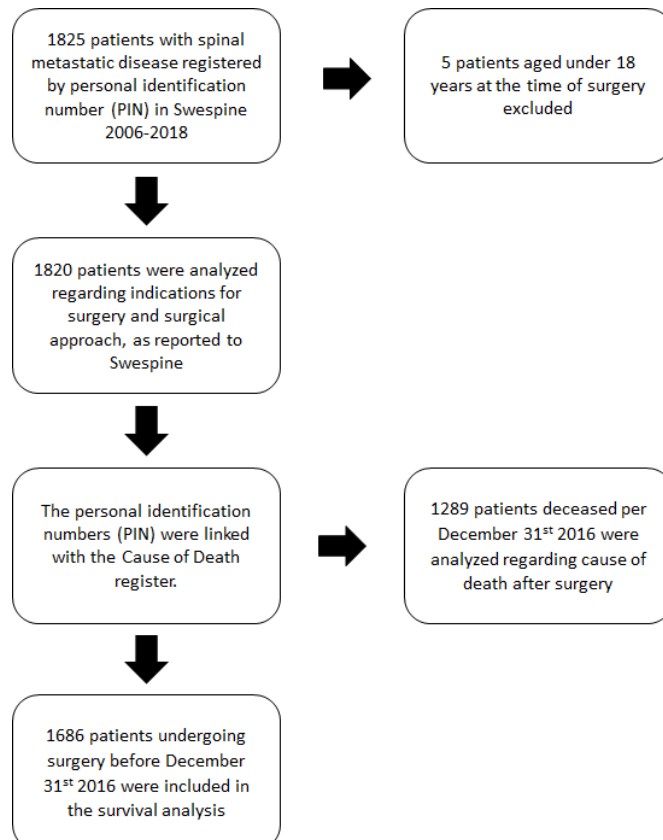

**Figure 1** Flowchart for inclusion and exclusion in the study.

due to metastatic spine disease. The study protocol follows RECORD guidelines.[8]

## Setting

All patients were treated in Sweden, a developed country with a national universal healthcare system.[9] There is virtually no private cancer care available, thus most of the included cases are covered by public healthcare institutions. The patients were treated at 19 different public hospitals, where 6 university hospitals accounted for 1646 (90%) of the cases.

Since reporting of healthcare quality is a legal requirement, most Swedish hospitals choose to report using the national healthcare quality registries. Swespine is the Swedish registry for spine surgery and has been in use since 1993. The module for spinal metastatic disease was introduced in 2006 and the registry as whole has a coverage (reflecting the spine surgery centres connected to the registry) of about 90% and a completeness (number of procedures entered in the registry) of 75% during the period.[10]

For deceased patients, the date of death and cause of death (CoD) are registered on the death certificate by the examining physician, or the pathologist when a post-mortem is performed. The certificates are entered into the Swedish CoD Register, where date of death and CoD are part of a death record. The register contains data from 1961 and onwards. The physician responsible for the patient is requested to report the CoD to the register.

If no CoD is reported, a reminder is eventually sent out. After reminders 1%–2% of the deaths still lack a cause of death and are then coded as 'Death NOS', R99.9 according to International Statistical Classification of Diseases, version ten (ICD-10).

As every individual residing in Sweden has a personal identification number (PIN), linkage of registries using the PIN is possible.

## Participants

Patients 18 years and older, undergoing surgery 1 August 2006 to 25 August 2018 due to spinal metastatic disease and registered in the spinal metastasis module of Swespine, were included (figure 1).

## Patient and public involvement

Patients or the public were not involved in the design, or conduct, or reporting, or dissemination plans of our research.

## Variables
### Survival

Survival after surgery was the primary endpoint and was calculated using the time difference from the date of surgery as reported in Swespine and the date of death from the CoD register. Patients surviving at least until the date of registry extraction were treated as censored cases. For the whole cohort (including patients alive as of 31 December 2016), the survival was estimated using the Kaplan-Meier method.

Due to the high level of missing data on tumour type in Swespine, we stratified data on tumour type from the CoD register and from Swespine separately, for comparison and validation of registry data.

### Type of tumour

The type of primary tumour is reported by the surgeon in Swespine as known or unknown at the time of surgery. If known, the surgeon can choose one of eight types to specify. The available choices are prostate, breast, renal, thyroid, lung, haematologic, gastrointestinal or other.

### Cause of death

The CoD register includes the CoD diagnosis according to ICD-10. If the reported main CoD was a neoplasm (ICD-10 codes C00-D48), this was considered the primary tumour in the analysis. In cases where multiple reasons of death were reported, the first neoplasm diagnosis (ICD-10 codes C00-D48) among the reported codes was used.

The quality of the Swedish CoD register is described as high with a completeness of 97%. While the autopsy rates have dropped in the last decades, the quality of the reporting was not necessarily negatively affected, thanks to better diagnostic tools.[11] A retrospective study on 5675 patients with prostate cancer published in 2009 reported an 86% correlation between the registered CoD and the CoD according to the reviewed records.[12]

## Indications for surgery

The surgeon reports the indication for surgery in Swespine. The possible indications are neurologic deficits, pain, progressing deformity or any combination of those.

## Neurologic function

The patient's neurologic function is reported according to the Frankel scale A–E, where A means a complete neurologic deficit and E means full motoric and sensory function.[13]

## Types of surgery

The surgeon reports the type of decompressive surgery (posterior/anterior and levels) and types of implants used (yes/no to posterior and anterior implants) to Swespine. If both questions regarding implants were answered 'no', we assumed that the patient was operated with decompression only.

In Swespine, the spinal levels of surgery are reported by the surgeon. The levels of decompression and stabilisation (if applicable) are reported, as well as the most proximal level of tumour mass addressed by the procedure. The most proximal level of tumour mass is used in this study to determine if the tumour mass is on cervical, thoracic or lumbosacral level. The decompression and/or stabilisation might cover more proximal levels than the reported level.

## Quality of life

The patients are requested to assess their HRQoL before surgery by filling in an EQ-5D form. EQ-5D is a tool measuring HRQoL in five dimensions and is widely used and validated in several medical fields.[14] Six weeks after surgery, the patient is requested to repeat the questionnaire and the values before and after surgery are reported to Swespine.

## Data sources/measurement

The spinal metastasis registry extract was provided by Swespine and sent to the Swedish National Board of Health and Welfare, where dates and CoD were linked with the spinal metastasis registry dataset. An anonymised

dataset was provided to the authors while the key remained with the agency.

## Bias

All patients in this cohort were selected to undergo surgery and probably had a better performance status than patients not selected for surgery. There is a likely attrition bias in reporting QoL, as patients with low performance status could be assumed to have less interest in filling out questionnaires, including the questionnaire 6 weeks postoperatively.

Registration of patient data in quality registries in Sweden is regulated by law and it is assumed that the patient agrees to participation, unless otherwise stated.[15] This reduces the risk of selection bias.

## Study size

The study size was determined by the number of patients with spinal metastatic disease registered in Swespine. No power calculations were made prior to the study.

## Quantitative variables

### Type of surgery

The surgeon-reported data on surgery includes type of surgery (anterior and/or posterior decompression, type of implant used) and spinal levels of surgery. The most proximal part of tumour mass addressed by the surgery is registered and this value is used to classify the metastases as cervical, thoracic or lumbosacral.

## Statistical methods

All calculations were made with the statistical software IBM SPSS Statistics V.26, including the survival estimations. The survival analysis was done with the Kaplan-Meier method. To test the statistical significance in survival between groups, the Breslow test was used.

## Data access and cleaning methods

The authors have full access to the Swespine database, and the anonymised dataset with the patients in the spinal metastases cohort. This dataset matches the dataset received from the CoD register and no further cleaning of cases has been made.

## Linkage

The authors were provided with an already-linked dataset. The linkage was performed by the National Board of Health and Welfare.

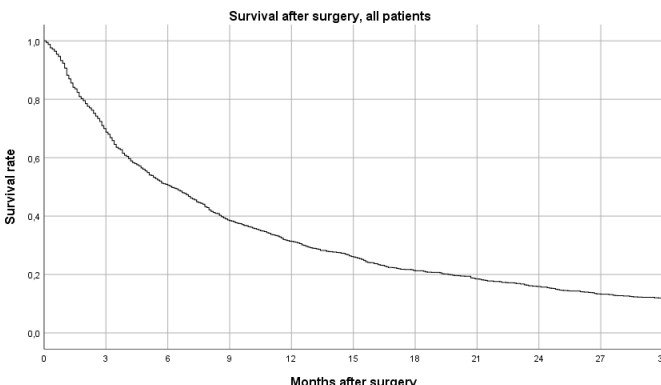

**Figure 2** Survival after surgery, all tumour types.

## RESULTS

### Participants

A total of 1820 adult patients were included in the analysis. The patients were treated at 19 different hospitals in Sweden. Six university hospitals accounted for 1646 (90%) of the cases.

**Table 1** Primary tumour as known before surgery reported to Swespine

| Type of primary tumour | n | % |
|---|---|---|
| Known | 773 | 42.5 |
| Unknown | 301 | 16.5 |
| Data missing | 746 | 41.0 |
| Total | 1820 | 100 |

**Table 2** Estimated survival after surgery for patients with known versus unknown primary tumour at the time of surgery, as reported to Swespine

| Primary tumour | Cases | Median estimated survival after surgery, months (95% CI) | Mean estimated survival after surgery, months (95% CI) |
|---|---|---|---|
| Known | 773 | 6.3 (5.4 to 7.2) | 12.0 (10.9 to 13.2) |
| Unknown | 301 | 5.3 (3.8 to 6.8) | 14.6 (12.0 to 17.2) |

## Descriptive data

The included patients were 18–95 years old at the time of surgery (mean age 67 years, 67% male).

## Outcome data

### Survival

The estimated median survival after surgery was 6.2 months (95% CI: 5.6 to 6.8) and the mean survival 12.2 months (95% CI: 11.4 to 13.1). As of 31 January 2016, 531 (29.2%) of the patients were alive and 1289 (70.8%) were deceased. Forty-nine per cent were dead within 6 months after surgery and 68.6% were dead within 12 months after surgery (figure 2).

### Cause of death

A total of 1253 of the deceased patients (97%) had a tumour diagnosis (ICD-10 codes C00-D48) listed as the main or contributing CoD, according to the CoD register. In 26 (2.0%) of the cases, no neoplasm diagnosis was reported as a contributing CoD. Ten (0.7%) patients had R99.9 (Death NOS) as CoD, indicating missing data.

### Survival per tumour type

In 773 (42.5%) cases, the tumour type was known before surgery according to Swespine, while it was stated as unknown in 301 (16.5%) cases. There was missing data regarding 746 (41.0%) cases (table 1).

The median estimated survival after surgery for patients with known primary tumour was 6.3 months and mean survival 12.0 months. For unknown primary tumour, median survival was estimated to be 5.3 months and mean survival 14.6 months. This difference was not statistically significant (p=0.68) (table 2).

Stratified estimated survival data for the three largest specified tumour groups (prostate, breast, lung) were calculated both from Swespine and the CoD register. A total of 429

patients had any of these tumour types registered in Swespine. The CoD register, which includes 1289 patients from the total cohort, had 679 patients with any of these diagnoses. There is an overlap between the registries, as cases might be registered correctly both in Swespine and the CoD register. On the other hand, patients still alive per 31 December 2016 are not included in the CoD register. Using the Breslow test, we found no significant differences in survival between the groups registered in Swespine and the groups with data from the CoD register, despite obvious sample size differences. Of the three sub-analysed tumour groups, patients with breast cancer had the longest estimated median and mean survival (table 3, figure 3).

### Indications for surgery

The most common indication for surgery was neurologic deficit, either on its own or in combination with other symptoms. Data on indication were missing in 41% of the cases (table 4).

### Neurologic function

The neurologic status before surgery according to the Frankel scale is reported to Swespine. Frankel level C was the most commonly reported level, while data were missing in 43 cases (table 5).

### Types of surgery

Posterior decompression was performed in 1592 (87.4%) cases, while 141 (7.7%) patients were decompressed anteriorly. Sixty-three (3.5%) patients were decompressed both anteriorly and posteriorly. Data regarding decompression were missing in 1.3% of the cases. It is unclear from the data if this means that no decompression was done (table 6).

**Table 3** Survival per tumour type from the three largest groups, as reported in the CoD register and Swespine, respectively

| Primary tumour | Cases in the CoD register | Median estimated survival after surgery, months (95% CI) | Mean estimated survival after surgery, months (95% CI) | Cases in Swespine | Median estimated survival after surgery, months (95% CI) | Mean estimated survival after surgery, months (95% CI) | P value |
|---|---|---|---|---|---|---|---|
| Prostate | 405 | 7.2 (6.2 to 8.2) | 11.5 (10.2 to 12.8) | 277 | 7.6 (6.4 to 8.9) | 12.6 (10.9 to 14.3) | 0.25 |
| Breast | 111 | 10.4 (5.2 to 15.6) | 17.7 (14.6 to 20.9) | 74 | 8.5 (5.4 to 11.6) | 17.7 (13.4 to 22.0) | 0.46 |
| Lung | 163 | 3.4 (2.4 to 4.4) | 7.6 (6.0 to 9.3) | 78 | 3.2 (2.6 to 3.8) | 6.8 (4.8 to 8.9) | 0.71 |
| Total | 679 | | | 429 | | | |

CoD, cause of death.

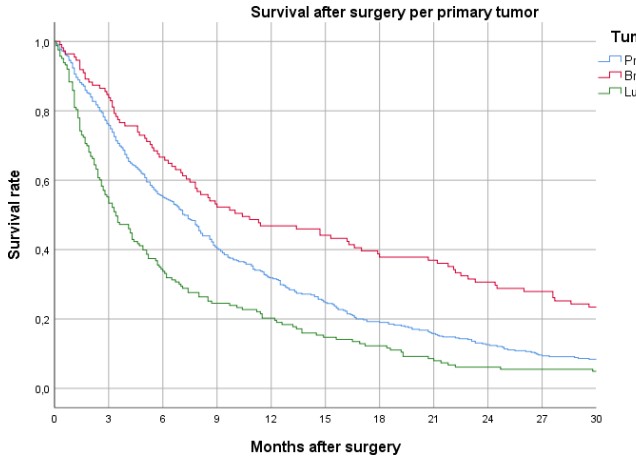

**Figure 3** Survival after surgery per primary tumour.

## Types of implants

The most common surgical procedure in this material was decompression combined with stabilisation. Posterior implants were used alone or in combination with anterior implants in 70.5% of the cases. In 378 cases, no implants were used at all. It is unclear whether missing data indicates that no implants were used or not. The type of implant (model/manufacturer) is reported to Swespine and the most common type used in this cohort was a posterior pedicle screw system (table 7).

## Spinal levels of surgery

Thoracic lesions were far more common than lumbosacral and cervical lesions. Th4 is the most common proximal level addressed by surgery in this material (figure 4).

## Quality of life

A total of 640 (35.2%) patients had a registered value for EQ-5D as assessed prior to surgery and 909 (49.9%) patients had EQ-5D data registered 6 weeks after surgery. The concordance of the two groups was low, as only 411 (22.6%) had data registered both before and after surgery. This subgroup had a higher mean EQ-5D level before surgery as well as after surgery (table 8).

## DISCUSSION
### Key results

This is one of the largest single-country cohorts with prospective baseline data, reliable survival data and fairly recently treated patients. The Swedish system with a unique PIN makes it possible to cross-reference and obtain data from several registries, which is one of the strengths of this study.

The median survival after surgery for metastatic spine disease in this cohort is above the levels recommended in predictive scoring systems where there is a time-based level of expected survival (usually 3 months), indicating that more patients could be eligible for surgery given the

**Table 4** Indications for surgery as reported to Swespine

| Indication(s) for surgery | n | % | Median estimated survival after surgery, months (95% CI) | Mean estimated survival after surgery, months (95% CI) |
|---|---|---|---|---|
| Neurologic deficit (A) | 684 | 37.6 | 5.8 (4.9 to 6.7) | 12.6 (11.2 to 13.9) |
| Pain (B) | 161 | 8.8 | 7.8 (5.8 to 9.8) | 16.4 (12.8 to 19.9) |
| Progressive deformity (C) | 19 | 1.0 | 6.5 (0.0 to 13.5) | 15.8 (3.2 to 28.3) |
| A+B | 154 | 8.5 | 5.3 (3.8 to 6.8) | 10.2 (8.0 to 12.4) |
| A+C | 11 | 0.6 | 2.8 (0.0 to 6.9) | 6.0 (1.2 to 10.7) |
| B+C | 20 | 1.1 | 5.3 (0.0 to 12.4) | 9.2 (3.3 to 15.2) |
| A+B+C | 31 | 1.7 | 11.6 (1.7 to 21.5) | 17.1 (10.1 to 24.1) |
| Data missing | 740 | 40.7 | | |
| Total | 1820 | 100 | | |

**Table 5** Neurologic status before surgery as reported to Swespine and estimated survival

| Frankel level | n | % | Median estimated survival after surgery, months (95% CI) | Mean estimated survival after surgery, months (95% CI) |
|---|---|---|---|---|
| A | 37 | 2.0 | 4.0 (2.3 to 5.7) | 9.4 (4.7 to 14.2) |
| B | 78 | 4.3 | 5.7 1.9 to 9.5) | 11.7 (7.6 to 15.6) |
| C | 430 | 23.6 | 5.2 (4.0 to 6.4) | 10.9 (9.4 to 12.4) |
| D | 298 | 16.4 | 6.9 (5.0 to 8.8) | 14.7 (11.7 to 15.7) |
| E | 192 | 10.5 | 7.8 (5.0 to 10.6) | 17.3 (13.8 to 20.9) |
| Data missing | 785 | 43.1 | | |
| Total | 1820 | 100 | | |

**Table 6** Types of decompression as reported to Swespine and estimated survival

| Type of decompression | n | % | Median estimated survival after surgery, months (95% CI) | Mean estimated survival after surgery, months (95% CI) |
|---|---|---|---|---|
| Posterior | 1592 | 87.5 | 5.4 (4.1 to 6.7) | 12.0 (11.2 to 12.9) |
| Anterior | 141 | 7.7 | 8.0 (4.0 to 12.0) | 18.2 (13.4 to 23.1) |
| Posterior and anterior | 63 | 3.5 | 9.4 (4.0 to 14.8) | 21.1 (12.7 to 29.4) |
| Data missing | 24 | 1.3 | | |
| Total | 1820 | 100 | | |

**Table 7** Types of implants used as reported to Swespine and estimated survival

| Type(s) of implant | n | % | Median estimated survival after surgery, months (95% CI) | Mean estimated survival after surgery, months (95% CI) |
|---|---|---|---|---|
| Posterior only | 1229 | 67.5 | 6.6 (5.8 to 7.4) | 12.3 (11.3 to 13.3) |
| Anterior only | 133 | 7.3 | 4.9 (1.4 to 8.4) | 16.4 (11.7 to 12.0) |
| Posterior and anterior | 55 | 3.0 | 11.4 (0.0 to 23.6) | 23.5 (13.6 to 33.4) |
| No implant | 378 | 20.7 | 4.1 (2.7 to 5.5) | 12.3 (6.9 to 17.6) |
| Data missing | 25 | 1.4 | | |
| Total | 1820 | 100 | | |

relatively long survival.[16] The estimated mean survival is almost twice as long as the estimated median survival (12.2 months vs 6.2 months) which can be explained by the minority of patients living several years after the surgical treatment. As previously shown by our group, the use of old prediction models may withhold patients from surgery as they tend to underestimate survival, and this study once again shows the need for reliable prediction tools.[17]

The Global Spine Tumour Study Group (GSTSG) has published several articles based on an international cohort of comparable size. While the patients in that cohort show a higher survival rate, 1 year after surgery, the

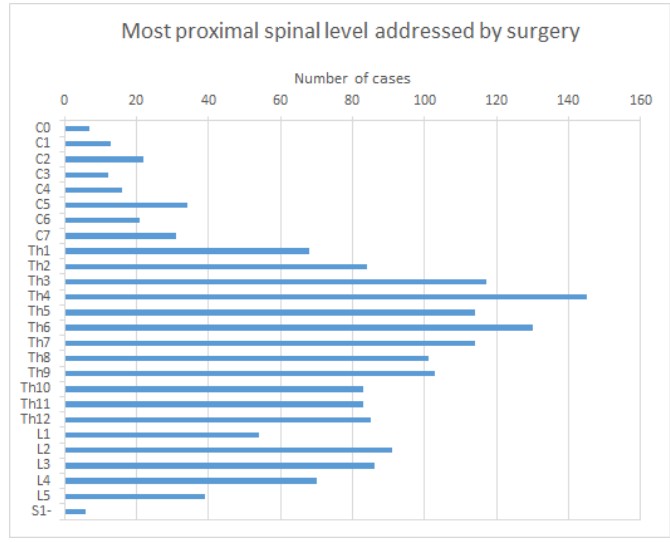

**Figure 4** Number of cases per proximal level addressed by surgery.

mean age is lower (60 years vs 67 years in our article) and the distribution of cancer types is different. Patients with breast cancer and lung cancer were both more common than prostate cancer in the GSTSG cohort and sarcomas were included as well. The heterogeneity between these cohorts complicates any comparative analysis.[3 18]

Not surprisingly, almost every patient with spinal metastatic disease in our cohort died of cancer rather than from other reasons. This emphasises the need to minimise complications after surgery in this fragile group.

Neurologic deficit was the most common indication for surgery reported in our article (38%), which is in line with the AOSpine Study published in 2015, where neurologic deficit was reported as the indication for surgery in 40% of the cases.[19] In our article, pain was the main indication for surgery in only 9% of the cases, while the AOSpine Study had 'intractable pain' as main indication in 39% of the cases. Unfortunately, the rate of missing data regarding indication for surgery is high (43%) in our article, which makes any conclusions uncertain. However, the AOSpine cohort is more recent than ours and we believe the difference illustrates the development in this field of spine surgery, where pain and instability are gaining more importance as indications. The Spine Instability Neoplastic Score, presented in 2010, has been validated as being correlated to patient-reported outcome after surgery which underlines the importance of pain and instability as indications for surgery.[20 21]

The focus of this study is on survival after surgery, which is an important factor when making treatment decisions. However, we recognise that the main goal of the treatment is not to prolong the life but rather to increase the HRQoL for the patient. Unfortunately, the amount

**Table 8** EQ-5D assessments and estimated survival in different subgroups, depending on data availability before and/or after surgery

| EQ-5D assessment | n | % | EQ-5D mean | SD | Median estimated survival after surgery, months (95% CI) | Mean estimated survival after surgery, months (95% CI) |
|---|---|---|---|---|---|---|
| Before surgery | 640 | 35.2 | 0.11 | 0.40 | 7.6 (6.6 to 8.6) | 14.6 (13.0 to 16.2) |
| Six weeks after surgery | 909 | 49.9 | 0.37 | 0.38 | 10.2 (9.0 to 11.4) | 16.0 (14.8 to 17.3) |
| Subgroup A assessed both before and 6 weeks after surgery | 411 | 22.6 | – | – | 11.4 (9.6 to 13.2) | 17.6 (15.6 to 19.7) |
| Subgroup A: before surgery | 411 | 100 | 0.14 | 0.39 | – | – |
| Subgroup A: 6 weeks after surgery | 411 | 100 | 0.38 | 0.39 | – | – |

Subgroup A had EQ-5D data both before and after surgery.

of missing data regarding HRQoL in the spinal metastasis module of Swespine is very high. Almost 8 out of 10 patients have missing data on HRQoL and any conclusions from this material should be drawn with caution. As expected, the mean EQ-5D in the group being assessed both before and after surgery was higher than in the other groups only answering one of the questionnaires. This subgroup increased the mean EQ-5D score with 0.24, far above the suggested minimally important difference which is around 0.10 for patients with cancer.[22]

Among those who assessed their HRQoL before surgery, the EQ-5D was low with a mean of 0.11. This level is comparable to the mean EQ-5D for dementia and lower than for palliative breast cancer and cerebral haemorrhage.[23]

The distribution of the metastases in the spine, where the thoracic spine accounts for the majority followed by the lumbar and cervical spines, is in line with other recent findings.[24 25]

### Limitations

The main limitation of this study is the absence of a nonsurgical control group and the lack of reliable reporting of adverse events. Swespine has a system for reporting adverse events in connection to the surgery, but finding late complications is a cumbersome task including manual assessment of multiple medical records and beyond the scope of this study. Other authors have concluded that the rate of adverse events after spine surgery is probably under-reported and we have no reason to believe that the situation is different in Sweden.[26]

Several outcomes in Swespine have a high level of missing data, which is another obvious limitation. With a completeness of the registry of around 75%, we can assume that there are several hundred cases during the period which are not entered into Swespine and thus not included in our study.

### Interpretation

Survival after surgery for spinal metastatic disease in this cohort is well above the recommended minimum and surgery might be considered in even more cases. Further research should emphasise prediction of survival and evaluation of HRQoL.

### Generalisability

The results of this study do not apply to all patients with spinal metastatic disease, given the heterogeneity between different cancer forms and the lack of a nonsurgical control group.

**Contributors** CC drafted the manuscript, made the statistical analyses in is the guarantor of the article. CO was involved in the study design and reviewed the manuscript. YR designed the study, wrote the ethical board application and reviewed the manuscript.

**Funding** The authors have not declared a specific grant for this research from any funding agency in the public, commercial or not-for-profit sectors.

**Competing interests** Dr Carrwik and Dr Robinson have nothing to disclose. Dr Olerud reports personal fees from Johnson & Johnson (paid speaker on course) outside the submitted work.

**Patient and public involvement** Patients and/or the public were not involved in the design, or conduct, or reporting or dissemination plans of this research.

**Patient consent for publication** Not applicable.

**Ethics approval** This study was approved by the Regional Ethical Review Board in Uppsala (no. 2012/133).

**Provenance and peer review** Not commissioned; externally peer reviewed.

**Data availability statement** Data are available upon reasonable request. Source data was accessed from Swespine and The National Cause of Death Register. Swespine is available to researchers subject to application (http://www.swespine. se) and the National Cause of Death Register is open-access, see https://www. socialstyrelsen.se/en/statistics-and-data/registers/

**ORCID iDs**
Christian Carrwik http://orcid.org/0000-0002-3092-8139
Yohan Robinson http://orcid.org/0000-0002-2724-6372

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
