## [Reviewer comments · BMJ Open]

ARTICLE DETAILS

TITLE (PROVISIONAL)	Survival after surgery for spinal metastatic disease: A nation-wide multi-registry cohort study
AUTHORS	Carrwik, Christian; Olerud, Claes; Robinson, Yohan

VERSION 1 – REVIEW

REVIEWER	Reiner, Anne Memorial Sloan Kettering Cancer Center
REVIEW RETURNED	17-Feb-2021

GENERAL COMMENTS	The authors report on survival following surgery for metastatic spine disease in 1820 adults from 2006-2016 using data from the Swedish national spine surgery register linked with the cause of death register. Survival was stratified by tumor type. Secondary outcomes reported were surgery indication, surgery type, and cause of death. Comments: 1. The sample size and date ranges are not consistent. Sometimes the population is 1820 (abstract), sometimes it is 1686 (those who had surgery before 2017), sometimes it is 1626 (Table 1). There is some attempt with the flowchart to clarify this, but it is not sufficient.2. The results are not consistently reported throughout. Some examples: In the abstract, posterior decompression and stabilization is performed in 70.5% of cases. This doesn't match Table 6. The flowchart and manuscript Survival section note that 1289 patients died, but in the manuscript Cause of Death section, only 1279 are accounted for.3. The reporting of survival by tumor type is not complete. The results are provided only for tumor type reported in the cause of death registry (Table 3 and Figure 3). The authors argue that this is because of the amount of missing data per Swespine but also note that this could underestimate survival time. I don't find this argument convincing. The authors already show in Table 2 that there is no difference in survival between those with known vs unknown Swespine tumor type (pvalue in manuscript 0.68). There are still 773 with known tumor type, a substantial sample size. Survival by tumor type per Swespine should also be reported.4. Indication for surgery, Frankel level, and type of decompression are all descriptively reported. Why not also report survival stratified by this information?5. There is information on QOL at surgery for 65% of patients but there is no reporting of this outcome even descriptively. Why? There is more data available for QOL than tumor type which is reported. Was there a survival difference in those with known versus unknown QoL at the two different timepoints it was collected? I still think it is worthwhile reporting change in QoL for
---

	the 23% who have it, even with the limitation noted of high level of missingness.
--	---

REVIEWER	Gauti Sigmundsson, Freyr Orebro universitet
REVIEW RETURNED	29-Aug-2021

GENERAL COMMENTS	Survival after surgery for spinal metastatic disease: A nation-wide multi-registry cohort study This is a descriptive study analysing the survival after surgery for spinal metastasis. The authors recruited the surgical patients from the Swedish Spine Register and linked the identified patients to the cause of death register. This is made possible by the unique personal identification number used in Sweden. The linkage of the databases was performed by the department of health and welfare. The aim of the study is to estimate the duration of survival of patients operated due to spinal metastasis and compare with other international cohorts of similar patients. The authors also describe the indications for surgery and survival based on the type of neoplasm (if known). General comments The manuscript is interesting and relevant for spine surgeons and oncologists. As with all studies employing health care databases there are issues of underreporting and missing variables. The authors make up for this to some extent by combing two databases. In the paper, the authors honestly describe the limitations of the study. The objective of surgery is palliative and survival is only one side of the coin. The other is health related quality of life during the remainder of life. Unfortunately, there are so many missing values in the SWESPINE regarding the health related quality of life that it precludes meaningful analysis. The authors do not elaborate as to the cause of non-responders but for the reader the authors thoughts on the issue may be interesting to know. Despite this I find the results from the study relevant. Specific comments New treatment modalities in oncology have drastically changed the survival of many cancer types. This has subsequently led to more focus on quality of life and less focus on neurological symptoms when selecting patients for surgery due to spinal metastasis. More focus is currently on surgery for pain. The spinal neoplastic instability score (SINS) is used for this. In the current study only 9% of the patients are operated for pain only and 1% for deformity. This leads me to believe that this study describes a historical cohort and the selection of patients for surgery may have changed drastically since these patients were selected for surgery. How does the indications for surgery compare to other recent
---

	cohorts in this respect. Please touch on this in the discussion. Furthermore, data regarding indication for surgery was missing in 41% of the cases. Please discuss the limitations of this. Survival time may also be underestimated in the study as only deceased were included. This information is important when interpreting the study. The difference between the median and mean survival time is 6 months. How should the reader interpret this? Is it meaningful to present both? Please elaborate. Can you estimate to which degree the study underestimates the number of surgeries for spinal metastases during the time period? Please discuss the issue. Page 7 (Quality of life) Line 19: fill in another form (repeat the questionnaire?) Page 8 (Bias) Surgical selection bias? Performance status is perhaps better than functional status in describing the bias as the Karnofsky performance scale is often employed in this setting. Attrition bias? Non-responders / missing / only deceased included Line 13: type of implant (Type of implant) Page 9 Line 2: What is meant by “cleaning of cases”. Are you referring to exclusion of cases? Line 15: 531 of the patients were alive and 1289 were deceased. Can you please add % values. Page 11 Line 5: Please add % values Discussion I believe the discussion can be elaborated on. Particularly, the results may be put in better perspective as modern oncological treatment has led to a paradigm change and the results from this study perhaps do not reflect current international practice with regards to the indications for surgery. The results presented in this paper may indicate that Swedish doctors exhibit restraint when selecting patients for surgery for spinal metastasis. This is relevant information, particularly in the face of modern oncological treatment. Perhaps this can be elaborated on in the discussion.
--	---

Reviewer: 1
Dr. Anne Reiner, Memorial Sloan Kettering Cancer Center

Comments to the Author:

The authors report on survival following surgery for metastatic spine disease in 1820 adults from 2006-2016 using data from the Swedish national spine surgery register linked with the cause of death register. Survival was stratified by tumor type. Secondary outcomes reported were surgery indication, surgery type, and cause of death.

Comments:

1. The sample size and date ranges are not consistent. Sometimes the population is 1820 (abstract), sometimes it is 1686 (those who had surgery before 2017), sometimes it is 1626 (Table 1). There is some attempt with the flowchart to clarify this, but it is not sufficient.

Table 1 is now corrected.

2. The results are not consistently reported throughout. Some examples: In the abstract, posterior decompression and stabilization is performed in 70.5% of cases. This doesn't match Table 6. The flowchart and manuscript Survival section note that 1289 patients died, but in the manuscript Cause of Death section, only 1279 are accounted for.

A table 7 is now added, describing the surgical techniques in more detail and explaining the 70.5% mentioned in the abstract, which refers to posterior stabilization (with or without decompression, corrected in the abstract).

The ten patients unaccounted for in the CoD section are now added (missing data).

3. The reporting of survival by tumor type is not complete. The results are provided only for tumor type reported in the cause of death registry (Table 3 and Figure 3). The authors argue that this is because of the amount of missing data per Swespine but also note that this could underestimate survival time. I don't find this argument convincing. The authors already show in Table 2 that there is no difference in survival between those with known vs unknown Swespine tumor type (pvalue in manuscript 0.68). There are still 773 with known tumor type, a substantial sample size. Survival by tumor type per Swespine should also be reported.

A very good argument. Survival by tumor type per Swespine is now added as Table 4 and there were no big differences despite the smaller sample size, underlining your point.

4. Indication for surgery, Frankel level, and type of decompression are all descriptively reported. Why not also report survival stratified by this information?

This has now been amended, and we added survival into the type of implant table as well. Please see the discussion part as well regarding these outcomes.

5. There is information on QOL at surgery for 65% of patients but there is no reporting of this outcome even descriptively. Why? There is more data available for QOL than tumor type which is reported. Was there a survival difference in those with known versus unknown QoL at the two different timepoints it was collected? I still think it is worthwhile reporting change in QoL for the 23% who have it, even with the limitation noted of high level of missingness.

Another very good point. QoL was out of scope for this study and the high amount of missing data made us focusing on other factors where we, thanks to the multi-registry design, had a lower level of missing data. After further review, we found preoperative EQ-5D data for even fewer patients (n 640 instead of n 1180). Please see table 9 for further info. The results are commented on in the discussion.

Reviewer: 2
Dr. Freyr Gauti Sigmundsson, Orebro universitet

Comments to the Author:

Survival after surgery for spinal metastatic disease: A nation-wide multi-registry cohort study

This is a descriptive study analysing the survival after surgery for spinal metastasis. The authors recruited the surgical patients from the Swedish Spine Register and linked the identified patients to the cause of death register. This is made possible by the unique personal identification number used in Sweden. The linkage of the databases was performed by the department of health and welfare.

The aim of the study is to estimate the duration of survival of patients operated due to spinal metastasis and compare with other international cohorts of similar patients.

The authors also describe the indications for surgery and survival based on the type of neoplasm (if known).

General comments

The manuscript is interesting and relevant for spine surgeons and oncologists. As with all studies employing health care databases there are issues of underreporting and missing variables. The authors make up for this to some extent by combing two databases. In the paper, the authors honestly describe the limitations of the study.

The objective of surgery is palliative and survival is only one side of the coin. The other is health related quality of life during the remainder of life. Unfortunately, there are so many missing values in the SWESPINE regarding the health related quality of life that it precludes meaningful analysis. The authors do not elaborate as to the cause of non-responders but for the reader the authors thoughts on the issue may be interesting to know.

Despite this I find the results from the study relevant.

Specific comments

New treatment modalities in oncology have drastically changed the survival of many cancer types. This has subsequently led to more focus on quality of life and less focus on neurological symptoms when selecting patients for surgery due to spinal metastasis. More focus is currently on surgery for pain. The spinal neoplastic instability score (SINS) is used for this. In the current study only 9% of the patients are operated for pain only and 1% for deformity. This leads me to believe that this study describes a historical cohort and the selection of patients for surgery may have changed drastically since these patients were selected for surgery. How does the indications for surgery compare to other recent cohorts in this respect. Please touch on this in the discussion. Furthermore, data regarding indication for surgery was missing in 41% of the cases. Please discuss the limitations of this.

This is a very good point and we elaborate on this in the updated discussion part, including a reference to the SINS.

Survival time may also be underestimated in the study as only deceased were included. This information is important when interpreting the study.

The survival is estimated with the Kaplan-Meier method and includes not only deceased patients.

The difference between the median and mean survival time is 6 months. How should the reader interpret this? Is it meaningful to present both? Please elaborate.

Both mean and median survival are normally presented in survival studies. We have now included an explanation as to why the estimated mean survival is almost twice as long as the estimated median survival.

Can you estimate to which degree the study underestimates the number of surgeries for spinal metastases during the time period? Please discuss the issue.

This is now discussed in the Limitations part.

Page 7 (Quality of life)

Line 19: fill in another form (repeat the questionnaire?)

This has now been changed.

Page 8 (Bias)

Surgical selection bias? Performance status is perhaps better than functional status in describing the bias as the Karnofsky performance scale is often employed in this setting.

Functional status has now been changed to performance status.

Attrition bias? Non-responders / missing / only deceased included

Attrition bias is probably an issue in the QoL analysis, where there is a high rate of missing data. This has been clarified. Please note that not only deceased patients are included in this study, except for the CoD analysis.

Line 13: type of implant (Type of implant)

Corrected.

Page 9

Line 2: What is meant by “cleaning of cases”. Are you referring to exclusion of cases?

We refer to the crossmatch between the Swespine metastasis module and the CoD register, where no cases were unaccounted for. We found no mismatch between the two registers.

Line 15: 531 of the patients were alive and 1289 were deceased. Can you please add % values.

Done.

Page 11

Line 5: Please add % values

Done.

Discussion

I believe the discussion can be elaborated on. Particularly, the results may be put in better perspective as modern oncological treatment has led to a paradigm change and the results from this study perhaps do not reflect current international practice with regards to the indications for surgery.

This subject is touched upon in the discussion part, where SINS also is discussed as described above.

The results presented in this paper may indicate that Swedish doctors exhibit restraint when selecting patients for surgery for spinal metastasis. This is relevant information, particularly in the face of modern oncological treatment. Perhaps this can be elaborated on in the

discussion.

This is an intriguing thought that we have discussed. As we have no control group and only survival data to backup this hypothesis, we only dare to conclude that "surgery might be considered in even more cases" (see interpretation). On a side note, there seem to be large differences between hospitals in Sweden regarding survival after surgery which might render an analysis for presentation in another forum.

VERSION 2 – REVIEW

REVIEWER	Reiner, Anne Memorial Sloan Kettering Cancer Center
REVIEW RETURNED	30-Sep-2021

GENERAL COMMENTS	Comments have been addressed. Thank you.
--

REVIEWER	Gauti Sigmundsson, Freyr Orebro universitet
REVIEW RETURNED	30-Sep-2021

GENERAL COMMENTS	I am pleased with the revision of this manuscript.
--